# ^89^Zr-Labeled Domain II-Specific scFv-Fc ImmunoPET Probe for Imaging Epidermal Growth Factor Receptor In Vivo

**DOI:** 10.3390/cancers13030560

**Published:** 2021-02-01

**Authors:** Elahe Alizadeh, Khan Behlol Ayaz Ahmed, Viswas Raja Solomon, Vijay Gaja, Wendy Bernhard, Amal Makhlouf, Carolina Gonzalez, Kris Barreto, Angel Casaco, C. Ronald Geyer, Humphrey Fonge

**Affiliations:** 1Department of Medical Imaging, College of Medicine, University of Saskatchewan, Saskatoon, SK S7N 0W8, Canada; elahe.alizadeh@queensu.ca (E.A.); khanbehlol@gmail.com (K.B.A.A.); vrajasolomon@gmail.com (V.R.S.); vijay.gaja@lightsource.ca (V.G.); amal.makhlouf@pharma.cu.edu.eg (A.M.); 2Department of Pathology and Laboratory Medicine, College of Medicine, University of Saskatchewan, Saskatoon, SK S7N 5E5, Canada; wlb302@mail.usask.ca (W.B.); cag948@mail.usask.ca (C.G.); kmb107@mail.usask.ca (K.B.); 3Department of Pharmaceutics and Industrial Pharmacy, Faculty of Pharmacy, Cairo University, Cairo 11562, Egypt; 4Centre for Molecular Immunology, Havana 11600, Cuba; casaco@cim.sld.cu; 5Department of Medical Imaging, Royal University Hospital (RUH), Saskatoon, SK S7N 0W8, Canada

**Keywords:** domain II-specific, EGFR, immunoPET imaging, monitoring response, ^89^Zr, anti-EGFR scFv-Fc fragment

## Abstract

**Simple Summary:**

Abundance of certain proteins such as epidermal growth factor receptor (EGFR) and their growth factors on cancer cells is in part responsible for their uncontrolled growth. Compounds that selectively bind to such proteins have diagnostic and/or therapeutic implications. EGFR has four binding domains (I-IV). Most anti-EGFR therapeutic antibodies bind to domain III. Compounds that bind to other domains have implications not only for diagnosis but also for monitoring therapy response. We describe the development of a diagnostic agent to be used with positron emission tomography (PET) that binds to domain II of EGFR. We developed ^89^Zr-8709-scFv-Fc antibody PET agent and evaluated its binding characteristics in cancer cells and mouse models. The presence of a domain III-binding antibody such as nimotuzumab did not inhibit the binding of ^89^Zr-8709-scFv-Fc, and vice versa. Therefore, ^89^Zr-8709-scFv-Fc PET/CT can be used for diagnosis and monitoring therapy response in the presence of a domain III-binding agent.

**Abstract:**

Epidermal growth factor receptor I (EGFR) is overexpressed in many cancers. The extracellular domain of EGFR has four binding epitopes (domains I- IV). All clinically approved anti-EGFR antibodies bind to domain III. Imaging agents that bind to domains other than domain III of EGFR are needed for accurate quantification of EGFR, patient selection for anti-EGFR therapeutics and monitoring of response to therapies. We recently developed a domain II-specific antibody fragment 8709. In this study, we have evaluated the in vitro and in vivo properties of ^89^Zr-8709-scFv-Fc (105 kDa). We conjugated 8709-scFv-Fc with the deferoxamine (DFO) chelator and radiolabeled the DFO-8970-scFv with ^89^Zr. We evaluated the binding of ^89^Zr-DFO-8709-scFv-Fc in EGFR positive and negative cell lines DLD-1, MDA-MB-231 and MDA-MB-435, respectively, and in mouse xenograft models. Simultaneously, we have compared the binding of ^89^Zr-8709-scFv-Fc with ^111^In-nimotuzumab, a domain III anti-EGFR antibody. DFO-8709-scFv-Fc displayed similar cell binding specificity as 8709-scFv-Fc. Saturation cell binding assay and immunoreactive fraction showed that radiolabeling did not alter the binding of 8709-scFv-Fc. Biodistribution and microPET showed good uptake of ^89^Zr-8709-scFv-Fc in xenografts after 120 h post injection (p.i). and was domain-specific to EGFR domain II. ^89^Zr-8709-scFv-Fc did not compete for binding in vitro and in vivo with a known domain III binder nimotuzumab. The results show that ^89^Zr-8709-scFv-Fc is specific to domain II of EGFR making it favorable for quantification of EGFR in vivo, hence, patient selection and monitoring of response to treatment with anti-EGFR antibodies.

## 1. Introduction

Epidermal growth factor receptor I (EGFR) is a 170 kDa transmembrane cell-surface glycoprotein which belongs to the ErbB family. EGFR and its family members play a variety of roles in the aberrant growth, oncogenesis and tumor progression in different cancers and cell types [1,2]. Overexpression of EGFR is known to drive a subset of aggressive cancers, including squamous cell head and neck [3], glioma [4], non-small cell lung, colorectal [5], ovarian [2,6], breast [7] and cervical cancers. In addition, a crosstalk of this receptor with many other receptors including those from its subfamily causes resistance to targeted therapies [8,9]. Structurally, EGFR consists of an extracellular ligand-binding, transmembrane, and intracellular domains. The extracellular binding domain has four subdomains, domains I–IV. Ligands of EGFR (epidermal growth factor, amphiregulin, transforming growth factor receptor α) bind to EGFR initiating a homo- and hetero-dimerization which results in downstream signal activation of the receptor(s) [2].

Ex vivo quantification of EGFR using biopsy samples are inherently flawed because of the intra-and inter-lesion heterogeneity of tumors. It is also known that EGFR expression of tumors changes over time [10,11]. In vivo assessment techniques have several advantages over current ex vivo methods, including measuring EGFR expression over the entire tumor volume rather than just a section of the tumor, assessing the biologic availability of EGFR in vivo, evaluating effects of therapy on EGFR expression, and quantifying EGFR expression of all lesions in real time.

EGFR overexpression in a wide range of cancers makes it a validated target for imaging and therapy using monoclonal antibodies. A few anti-EGFR monoclonal antibodies such as cetuximab, panitumumab, necitumumab and nimotuzumab are approved for treatment in different countries. These antibodies bind to domain III of EGFR [12,13,14,15]. Some of them have been radiolabeled with different isotopes for use as diagnostics. The clinical utility of ^89^Zr-cetuximab PET/CT in selecting patients that would benefit from treatment using an EGFR targeted monoclonal antibody was recently demonstrated in colorectal cancer patients by Menke-van der Houven van Oordt et al. [16]. Only patients with positive ^89^Zr-cetuximab PET/CT lesions showed clinical benefit with the therapeutic monoclonal antibody cetuximab. ^89^Zr-cetuximab and other domain III imaging agents can only be used for diagnosis and patient selection. They cannot be used to monitor response to treatment using anti-EGFR monoclonal antibodies, since the imaging probe will compete for binding with the therapeutic antibody.

Therefore, there is strong interest to develop novel imaging agents that bind to epitopes/domains that are different from that of the therapeutic agent. A domain II binder can find applications in patient selection, diagnosis and monitoring of response to anti-EGFR treatments. For the first time, Miersch et al. recently engineered a domain II (8709-scFv-Fc) specific anti-EGFR antibody using phage display [1]. Anti-EGFR antibody 8709-scFv-Fc inimitably binds to domain II of EGFR with a high affinity and does not interfere with the binding of clinically available anti-EGFR therapeutic antibodies e.g., cetuximab [17]. The expression, purification and optical imaging of the 8709-scFv-Fc antibody fragment has been published [17].

Here, we have evaluated the in vitro and in vivo properties of 8709-scFv-Fc fragment radiolabeled with ^89^Zr via a deferoxamine (*p*-SCN-Bn-deferoxamine) chelator in EGFR overexpressing cell lines and mouse xenografts and controls. We present the in vitro cell binding characteristics as well as in vivo biodistribution, pharmacokinetics and microPET imaging in tumor-bearing mice. Our results show that ^89^Zr-8709-scFv-Fc has the potential to be used for diagnosis, patient selection and treatment monitoring of EGFR-positive cancers.

## 2. Materials and Methods

### 2.1. Characterization of Antibody

Anti-EGFR antibody 8709-Fab was identified by phage display and converted to an scFv-Fc (8709-scFv-Fc) its expression, purification and fluorescent imaging has been published [17].

### 2.2. Cell Lines and Xenografts

EGFR positive DLD-1 (RRID:CVCL_0248) colorectal cancer, breast cancer MDA-MB-231 (RRID:CVCL_0062), and EGFR negative human melanoma MDA-MB-435 (RRID:CVCL_0417) cell lines were obtained from ATCC (Rockville, MD, USA). MDA-MB-435 cells are derived from M14 melanoma cells. Cells were propagated by serial passage in MEM/EBSS medium, supplemented with 10% fetal bovine serum (Biochrom, Sigma-Aldrich, St. Louis, MO, USA) at 37 °C in a humidified atmosphere of 5% CO_2_. All human cell lines have been authenticated using STR profiling within the last three years. All experiments were performed with mycoplasma-free cells. 

All animals used in imaging experiments were cared for and maintained under the supervision and guidelines of the University of Saskatchewan Animal Care Committee (UACC). All animal studies were approved by UACC in accordance with the guidelines set forth on the Use of Laboratory Animals (protocol # 20170084) [18]. Female CD-1 nude mice were obtained from Charles River Canada (St-Constant, Quebec) at 4 weeks of age and housed in a 12 h light, 12 h dark cycle in a temperature and humidity-controlled vivarium. Animals had libitum access to diet (Lab Diet, St. Louis, MO, USA) and water [18]. After one week of acclimatization, mice were subcutaneously injected with a suspension of 5–10 × 10^6^ DLD-1, MDA-MB-231 or MDA-MB-435 cells in 100 μL of a 1:1 mixture of serum-free MEM/EBSS medium (HyClone Laboratories, Logan, UT, USA) and Matrigel matrix basement membrane (Discovery Laboware, Inc. Bedford, MA, USA) at the hind limb of each mouse. Xenograft was located in the right thigh of the hind leg of the mouse and tumor growth was followed with caliper measurements.

### 2.3. Conjugation of Antibodies with Bifunctional Chelators

Conjugation of 8709-scFv-Fc (domain I/II binder) was performed as reported earlier [19]. The excess unreacted chelator was removed by centrifugation using an Amicon Ultra-10 K (Burlington, MA, USA) molecular filtration device. The solution of DFO-8709-scFv-Fc was aliquoted and stored at −80 °C until further use after quality control. 8709-scFv-Fc and DFO-8709-scFv-Fc purities were determined using SEC-HPLC (Waters 2796 Bioseparations Module, Waters 2487 Dual λ Absorbance Detector, XBridge BEH 200A SEC 3.5 µm 7.8 × 300 mm column, Waters Corporation, Milford, MA, USA). The UV-Detector was set at 220 and 280 nm and the solvent system was PBS at a flow rate of 0.6 mL/min. Nimotuzumab (domain III binder) was conjugated with DOTA for labeling with ^111^In following lab protocol [18].

The analysis of MW and purity of conjugated antibody was performed on an Agilent 2100 Bioanalyzer using Agilent High Sensitivity Protein 250 Kit using the manufacturer’s protocol. The size and relative peak area were calculated using Agilent 2100 Expert software (Agilent Technologies, Santa Clara, CA, USA, cat # 5067-1575).

Binding kinetics between the antibodies and recombinant monomeric EGFR were measure with ForteBio Octet RED384 (PALL Corporation, NY, USA). Antibodies were immobilized on anti-human FAB-CH1 sensors (18-5104, Forte Bio) according to manufacture’s instructions. The equilibrium dissociation constant (K_D_) was obtained using a 1 to 1 binding model with global fitting. Data analysis and curve fitting was performed using data analysis software 7.1.0.33 (Forte Bio, Santa Clara, CA, USA). Saturation binding flow cytometry was performed as described previously [17].

### 2.4. Production and Characterization of ^89^Zr-Oxalate

^89^Zr-oxalate was produced at the Saskatchewan Centre for Cyclotron Sciences (SCCS), University of Saskatchewan. Yttrium coated (10 mm diameter and 200 µm thickness) on the coin shaped niobium (24 mm diameter and 1 mm thickness) target supplied by ACSI, was irradiated with an incident beam energy of 17.8 MeV providing a degraded transmitted energy through an aluminum degrader of 12.8 MeV and a current of 40 µA for 2 h on an in-house beam-line on TR-24 cyclotron to produce ^89^Zr via the ^nat^Y(p, n)^89^Zr reaction. During irradiation the target was cooled on the frontal side by Helium gas and on the back side by chilled water. After the irradiation, the target left on the target station for 2–3 h to allow decay of short-lived isotopes. Then target was released in the lead pig and transported to the hot cells in the lead shielded cart.

^89^Zr was separated as ^89^Zr-oxalate from the irradiated yttrium coin as described by Queern et al. [20]. Briefly, to a dissolution vessel charged with the irradiated target 2 N trace metal HCl (4 mL) was added slowly. The resulting solution was warmed to 80 °C for 20 min and cooled to room temperature before loading onto a pre-conditioned column of hydroxamate resin (100 mg). A dissolution vessel and target body rinsed with 2 N trace metal HCl (2 mL) and loaded onto hydroxamate resin. Then hydroxamate resin was rinsed with 2N HCl (14 mL) followed by traceselect water (10 mL). Finally, ^89^Zr was eluted with 1.5 mL of 1.0 M Oxalic acid

^89^Zr-oxalate was characterized for activity amount, half-life, radionuclidic purity, elemental impurities and specific activity. The final activity of purified ^89^Zr-oxalate was measured in the Capintec dose calibrator (CRC-55t PET). Specific activity was determined using a published method by Holland et al. [21] and the specific activity was found to be in the range of 673-1161 MBq/µmol. The ^89^Zr-oxalate solutions were tested for radionuclidic identity and purity using a high purity germanium (HPGe) detector (Ortec, Oak Ridge, TN, USA). The radiochemical purity was 99.99% and the identity was confirmed by the presence of 909 keV and 511 keV peaks. The ICP-MS analysis of ^89^Zr-oxalate solutions were performed at Saskatchewan Research Council, Saskatoon and elemental impurity levels were found to be always below 30 ppm and all batches met the QC release specifications.

### 2.5. Radiolabeling and Characterization of ^89^Zr-8709-scFv-Fc and ^111^In-Nimotuzumab

Radiolabeling of DFO-8709 conjugate with ^89^Zr-oxalate was performed as per the standard lab protocol [19]. After radiolabeling the reaction mixture was cooled to room temperature and an aliquot was injected onto SEC-HPLC (the same HPLC and solvent system as above) for radiochemical quality control and radiolabeling efficiency determination. Oxalate, HEPES and other impurities (less than 30 kDa) were removed from the crude reaction mixture by single centrifugation in spin columns (Amicon Ultra-4 Centrifugal Filter 30K NMCO, 4 mL, EMD Millipore, Burlington, MA, USA) and sterilized by filtration with 0.22 µm hydrophilic PTFE membrane filter (Ultrafree-CL Centrifugal filter 4 mL, Millipore). The final solution was formulated in PBS. The recovery of the radiolabeled product was 60% for ^89^Zr-DFO-8709-scFv-Fc with a radiochemical purity of > 95% was used for in vitro and in vivo experiments.

Radiolabeling of DOTA-nimotuzumab conjugate with ^111^InCl_3_ was performed as per the standard lab protocol [18]. After labeling, the reaction was monitored using iTLC strip with 100 mM sodium citrate buffer (pH 5.0) as the mobile phase and analyzed using ScanRam (LabLogic, Brandon, FL, USA). ^111^In-labeled conjugates were purified using Amicon Ultra-4 centrifugal filters (10K, EMD Millipore) with PBS. The purity of the radiolabeled immunoconjugates were determined using size exclusion radio-HPLC and iTLC. A radiochemical purity (RCP) of more than 95% was considered good for in vitro and in vivo experiments.

### 2.6. In Vitro Binding

The in vitro saturation binding, immunoreactivity and domain specific binding of ^89^Zr-DFO-8709-scFv-Fc was determined in EGFR positive DLD-1 cells. A saturation radioligand binding assay was performed by incubating 0.5 million cells with increasing concentrations of radioimmunoconjugates (0.2–95 nmol/L in 100 μL PBS) for 4 h at 4 °C. Non-specific binding (NSB) was determined in a similar assay but in the presence of a 50-fold molar excess of unlabeled 8709-scFv-Fc (relative to the highest concentration of the radioimmunoconjugates). A non-linear regression analysis with one-site binding equation was used to determine K_D_ using GraphPad Prism 6 (GraphPad Software, La Jolla, CA, USA). The immunoreactive fraction of the radioimmunoconjugate was determined as described in Lindmo et al. [22].

Domain II specific binding was determined by a competitive binding experiment. Briefly, one million DLD-1 cells were transferred in 3 vials and incubated with ^89^Zr-8709-scFv-Fc, but with slight modifications. The first vial was incubated only with ^89^Zr-8709-scFv-Fc, second vial was initially incubated with cold 8709-scFv-Fc whose concentration was 50× the concentration of radiolabeled ^89^Zr-8709-scFv-Fc. After 2 h of pre-incubation with unlabeled 8709-scFv-Fc, ^89^Zr-8709-scFv-Fc was added to these cells. The third vial was incubated with unlabeled nimotuzumab whose concentration was 50 × the concentration of radiolabeled ^89^Zr8709-scFv-Fc. After 2 h pre-incubation with nimotuzumab, ^89^Zr-8709-scFv-Fc was added to these cells. After incubation, the cells were centrifuged at 1200 rpm and supernatant was collected separately. This process was repeated 3 times to ensure complete removal of unbound ^89^Zr-8709-scFv-Fc. After centrifugation the pellet was re-dissolved in PBS and samples were measured using gamma counter. Experiments were performed in triplicate using three different concentrations (50, 5 and 0.5 nM) of ^89^Zr-8709-scFv-Fc.

### 2.7. Biodistribution and Pharmacokinetics

Biodistribution studies were performed in athymic CD-1 nude mice bearing EGFR positive DLD-1, and EGFR negative MDA-MB-435 (control) xenografts (*n* = 3/group). Each mouse received a dose of with 10 ± 0.2 MBq of ^89^Zr-8709-scFv-Fc (20 µg, specific activity of 0.5 MBq/µg). Animals were sacrificed at 24 and 168 h post injection (p.i.) and different organs were harvested (blood, lungs, liver, spleen, kidneys, bladder, muscle, heart, brain, bones, thymus, pancreas, stomach, small, and large intestine) for biodistribution. The activity in the organs was measured using a gamma counter (2480 Perkin Elmer, Waltham MA, USA) and the tissue uptake was expressed as percentage of injected activity per gram (% IA/g).

The pharmacokinetics was determined in healthy Balb/c mice (*n* = 5/group). The animals were injected with 10 ± 0.0.1 MBq of the ^89^Zr-8709-scFv-Fc via a tail vein and blood was collected at various time points from a saphenous vein in heparinized capillary tubes. Activity in the capillary tube was measured using a gamma counter and the blood volume was determined using a digital caliper, assuming the internal diameter of the tube. Radioactivity in blood activity was expressed as % IA/mL). All relevant pharmacokinetic parameters were determined using an exponential decay curve fitting from sigma plot using GraphPad Prism 6.

### 2.8. EGFR Domain Specific Imaging Using ^111^In-Nimotuzumab and ^89^Zr-8709-scFv-Fc

In vivo evaluation of domain III binding of ^111^In-nimotuzumab was studied with two groups of mice (*n* = 3), one group bearing EGFR positive DLD-1 xenografts and other bearing EGFR negative MDA-MB-435 xenografts. Mice were injected with 15 ± 0.2 MBq of ^111^In-nimotuzumab (0.5 MBq/µg) via tail vein injection. Imaging was performed in mice injected with EGFR positive DLD-1 and EGFR negative MDA-MB-435 (control) xenografts. Imaging was performed at three different time points 24 h, 48 h, and 120 h p.i.

In vivo evaluation of domain II binding of ^89^Zr-8709-scFv-Fc was also studied with two groups of mice (*n* = 3), one group bearing EGFR positive DLD-1 xenograft and other bearing EGFR negative MDA-MB-435 xenograft. Mice were injected with 9 ± 0.2 MBq of ^89^Zr-8709-scFv-Fc via a tail vein. Imaging was performed at three different time points 24 h, 48 h, and 120 h p.i.

To understand site-specific domain II binding of ^89^Zr-8709-scFv-Fc, one group (*n* = 3) of mice were initially injected with 250 µg of non-radiolabeled nimotuzumab to block the domain III of EGFR leaving domain II available for ^89^Zr-8709-scFv-Fc binding.

Simultaneously imaging of domain III and domain II was also studied following simultaneous injection of ^89^Zr-8709-scFv-Fc and ^111^In-nimotuzumab. Two groups of mice (*n* = 3) one group bearing EGFR positive DLD-1 xenograft and other bearing EGFR negative MDA-MB-435 xenograft were simultaneously injected with 9 ± 0.2 MBq of ^89^Zr-8709-scFv-Fc and 15 ± 0.2 MBq of ^111^In-nimotuzumab via tail vein injection.

To confirm domain II specificity, groups of mice (*n* = 3) bearing EGFR positive DLD-1 or MDA-MB-231 xenograft was initially injected with 250 ug (therapeutic dose) of unlabeled nimotuzumab to block domain III, followed by simultaneous injection with 9 ± 0.2 MBq of ^89^Zr-8709-scFv-Fc and/or 15 ± 0.2 MBq of ^111^In-nimotuzumab and microPET/SPECT/CT imaging.

PET/CT images were acquired at 24 h, 48 h, and 120 h p.i. using the Vector^4^CT scanner (MILabs B.V., Utrecht, The Netherlands). PET scans were acquired in a list-mode data format with a high-energy ultra-high resolution (HE-UHR-1.0 mm) mouse/rat pinhole collimator. Corresponding CT scans were acquired and images were reconstructed using a pixel-based order-subset expectation maximization (POS-EM) algorithm that included resolution recovery and compensation for distance-dependent pinhole sensitivity and were registered on CT and quantified using PMOD 3.8 software (PMOD Technologies Ltd., Zurich, Switzerland). SPECT/CT images were acquired using collimator and detector settings as described previously [19].

Tracer uptake was expressed as percentage injected activity (% IA) per cc of tissue volume (% IA/cc). All quantification data was reported as mean ± standard deviation within one animal study group. After imaging all mice were sacrificed for biodistribution. Activity in the organs was analyzed as percent injected activity per gram (% IA/g).

### 2.9. Statistical Analysis

All data was expressed as the mean ± SD or SEM of at least three independent experiments. Statistical comparisons between the experimental groups were performed by a t-test. Graphs were prepared and *p* values were calculated using GraphPad Prism (version 5.03). *p* values of less than 0.5 were considered significant.

### 2.10. Data Availability

The data that support the findings of this study are available from the corresponding author upon reasonable request.

## 3. Results

### 3.1. Conjugation and Quality Control of Immunoconjugates

8709-scFv-Fc was conjugated to *p*-SCN-Bz-deferoxamine (DFO) resulting in 8709-DFO (Figure 1a). The resulting 8709-DFO was >99% pure, conjugation increased the molecular weight of the parent compound from 119 kDa to 120.5 kDa representing the incorporation of an average of 2 DFO molecules per antibody (Figure 1b). Binding to recombinant human EGFR was 1.5 fold worse when conjugated to DFO (KD_BLI_^(8709)^ = 55 ± 1 nM versus KD_BLI_^(8709-DFO)^ = 83 ± 10 nM) (Figure 1c). The binding constant to DLD-1 cells was not significantly affected by conjugation with DFO (KD_flow_^(8709)^ = 72 ± 5 nM, KD_flow_^(8709-DFO)^ = 97 ± 15 nM, *p*-value > 0.5) (Figure 1d). 8709 bound the control cell-line MDA-MB-435 with less than 3 ± 2% (Figure 1d).

### 3.2. Radiolabeling of ^89^Zr-DFO-8709 ScFv-Fc and ^111^In-Nimotuzumab

Radiolabeling of 8709-scFv-Fc with ^89^Zr and nimotuzumab with ^111^In was performed as per the standard laboratory protocol [18,19]. Radiolabeling was monitored after 30 min using iTLC. Radiochemical yield of >95% was obtained for ^89^Zr-8709-scFv-Fc within 2 h. The radiochemical purity for ^89^Zr-8709-scFv-Fc was >98% as observed by iTLC. Purity of radiolabeled conjugates was analysed using SEC-HPLC (Appendix A).

### 3.3. In Vitro Binding

Effect of radiolabeling on binding of radioimmunoconjugates to EGFR was analysed using saturation binding assay towards EGFR positive DLD-1 cells. It was observed that ^89^Zr-8709-scFv-Fc displayed good specific binding (Appendix A) which was evident from dissociation constant K_D_ value of 34.39 ± 4.02 nM. ^111^In-nimotuzumab also had good binding affinity towards EGFR positive DLD-1 with a K_D_ value of 14 nM. The immunoreactive fraction assay was performed to determine the fraction of ^89^Zr-8709-scFv-Fc that binds to DLD-1 cells. The immunoreactive fraction of ^89^Zr-8709-scFv-Fc was 0.73. A similar value was reported for ^111^In-nimotuzumab by our group [18].

To confirm the specific binding of ^89^Zr-8709-scFv-Fc to domain II of EGFR, competitive binding experiment was performed. ^89^Zr-8709-scFv-Fc specifically bound to domain II (Figure 2). In the presence of cold 8709-scFv-Fc no binding was observed, while in the presence of cold nimotuzumab (domain III binder), there was no reduction of ^89^Zr-8709-scFv-Fc binding confirming specificity of domain binding of ^89^Zr-8709-scFv-Fc.

### 3.4. Biodistribution and Pharmacokinetics

Biodistribution of ^89^Zr-8709-scFv-Fc and ^111^In-nimotuzumab was analysed at 24 and 168 h p.i. (Figure 3) in mice bearing xenografts with high EGFR expression (DLD-1) and an EGFR negative control (MDA-MB-435). There was no significant difference in tumor uptake of ^89^Zr-8709-scFv-Fc at 24 h (5.1 ± 1.6 %IA/g) and at 168 h (4.3 ± 1.4 %IA/g) (Figure 3a). Uptake of ^89^Zr-8709-scFv-Fc in control xenograft MDA-MB-435 was less (2.5 ± 0.6 %IA/g) indicating specificity of 8709-scFv-Fc towards EGFR (Figure 3b). Tumor uptake of ^111^In-nimotuzumab followed a similar trend with high uptake (12.4 ± 1.3 %IA/g) in EGFR positive DLD-1 xenograft (Figure 3c) and less uptake (3.4 ± 1.1 %IA/g) in EGFR negative MDA-MB-435 xenograft (Figure 3d). Tumor-to-blood ratios for ^89^Zr-8709-scFv-Fc was 0.4 and 3.1 in DLD-1 xenograft at 24 h and 168 h post injection, respectively, and was 0.9 and 4.3 in the same xenograft at 24 h and 168 h post injection, respectively for ^111^In-nimotuzumab.

^89^Zr-8709-scFv-Fc exhibited a bi-phasic clearance with a fast (distribution) half-life t_1/2α_ of 1.5 ± 1.2 h and a slow clearance t_1/2β_ of 134.1 ± 1.3 h (Appendix A and Table 1).

### 3.5. MicroPET/SPECT/CT Imaging Using ^111^In-Nimotuzumab and ^89^Zr-8709-scFv-Fc

MicroSPECT/CT imaging confirms the specific binding of ^111^In-nimotuzumab to EGFR (Figure 4a). The tumor-to-muscle ratio was found to increase from 4.8 ± 1.73 to 6.6 ± 2.20 by the end of 120 h. Tumor uptake in mice bearing control MDA-MB-435 xenografts was low (4.0 ± 1.44%IA/cc) at the end of 120 h (Figure 4b) which confirms specific uptake ^111^In-nimotuzumab to EGFR positive DLD-1 xenograft model. Tumor uptake of ^111^In-nimotuzumab was blocked when mice were pre-injected with a therapeutic dose of unlabeled nimotuzumab prior to SPECT imaging with the tracer (Figure 4c). Mice bearing DLD-1 xenografts, had high tumor uptake (14.2 ± 6%IA/cc) at 120 h p.i. (Figure 4d).

Binding of ^89^Zr-8709-scFv-Fc to domain II of EGFR was studied using PET imaging at different time points 24, 48, and 120 h p.i. (Figure 4 and Figure 5). Tumor uptake of ^89^Zr-8709-scFv-Fc gradually increased from 4.2 ± 0.7 % at 24 h p.i. to 6.0 ± 0.6 %IA/cc at 48 h (Figure 5a). Groups of mice pre-injected with unlabeled nimotuzumab before the injection of ^89^Zr-8709-scFv-Fc also showed similar tumor uptake at 24 h and 120 h p.i. (Figure 5b), confirming that a blocking dose of unlabeled nimotuzumab (domain III binder) did not alter the binding of ^89^Zr-8709-scFv-Fc (domain II binder) to EGFR. A similar trend in tumor uptake observed in DLD-1 xenograft was seen in EGFR positive MDA-MB-231 which has medium EGFR expression (Figure 5c,d). Mice bearing control MDA-MB-435 xenografts displayed lower tumor uptake of 3.1 %IA/cc at the end of 120 h (Figure 5e). Quantification of this uptake is presented in the supplementary section (Figure 5f and Appendix A) and shows no statistical difference.

Simultaneous targeting of domain III and domain II of EGFR was studied by dual tracer imaging following simultaneous administration of ^111^In-nimotuzumab and ^89^Zr-8709-scFv-Fc in mice bearing EGFR positive DLD-1 xenograft. Tumor uptake was 9.3 %IA/cc and 6.5 %IA/cc at 120 h p.i. for ^111^In-nimotuzumab and ^89^Zr-8709-scFv-Fc, respectively (Figure 6a–c). These values were not significantly different from those observed with single tracer imaging studies. Tumor-to-normal organ ratios for ^89^Zr-8709-scFv-Fc are presented (Appendix A).

## 4. Discussion

Molecular imaging agents such as ^89^Zr-cetuximab and ^89^Zr-panitumumab have been developed and evaluated in clinical trials [16]. However, because these agents bind to the same epitope (mostly domain III) as the anti-EGFR therapeutic antibody, they cannot be used to monitor response to treatment of the same antibody. Approved anti-EGFR antibodies e.g., cetuximab, panitumumab, nimotuzumab and more recently necitumumab bind to domain III leaving the dimerization loop of EGFR (domain II) available for binding to other molecules. Therefore, a monoclonal antibody that binds to a domain other than domain III is needed for comprehensive diagnosis, monitoring of response to anti-EGFR antibody therapy and potentially as a biparatopic anti-EGFR therapeutic agent. For the first time we have developed an immunoPET agent that binds to epitope II of EGFR.

Using phage display we have screened and engineered antibodies and antibody fragments that bind to domain II of EGFR [1]. We recently reported a near-infrared (NIR) fluorescently labeled IRDye800CW-8909 antibody and antibody fragments of this antibody [17]. 8907-scFv-Fc emerged as the best fragment for imaging domain II of EGFR [17]. The expression of 8907-scFv-Fc (50 mg/mL) was at least four-fold better than the other fragments and IgG. In addition, NIR showed that tumor uptake and target to organ ratios of 8907-scFv-Fc was better than all other fragments including the IgG. Here, we report the initial evaluation of a ^89^Zr-labeled 8907-scFv-Fc fragment of the antibody fragment for potential use in non-invasive diagnosis, patient selection and monitoring of response to anti-EGFR monoclonal antibody treatment by PET.

Conjugation of 8709-scFv-Fc resulted in an average of two molecules of DFO attached to 8709-scFv-Fc which was evident from bioanalyzer. Bioanalyzer results show that there was a slight difference in the molecular weight of 8709-scFv-Fc and DFO-8709-scFv-Fc. Flow cytometry data suggests that, binding affinity of DFO-8709-scFv-Fc was not significantly (72 nM vs 97 nM for 8709-scFv-Fc and DFO-8709-scFv-Fc, respectively in DLD-1 cells) altered by conjugation, indicating conjugation and chemicals used in the reaction did not alter the characteristics of the antibody fragment.

The strategy to develop an antibody imaging agent that binds to a different epitope from the therapeutic antibody has already been investigated using anti-HER2 antibodies pertuzumab (domain II) and trastuzumab (domain IV) [23,24,25]. Pertuzumab (Perjeta) and trastuzumab (Herceptin) are approved for treating HER2 positive breast cancer [26]. A number of authors have developed radiolabeled derivatives of pertuzumab for imaging and monitoring of response to trastuzumab [23,25]. Marquez et al. [23] evaluated ^89^Zr-pertuzumab in mice bearing HER2 positive xenografts with or without the administration of unlabeled trastuzumab. It was shown that the administration of unlabeled trastuzumab 5–60 min before the administration of ^89^Zr-pertuzumab led to an increase in tumor uptake of ^89^Zr-pertuzumab compared to mice without pre-administration of unlabeled trastuzumab suggesting treatment with unlabeled trastuzumab may affect the quantification of HER2, despite the fact that both antibodies bind to unique epitopes of the receptor. This observation remains unclear and should be interpreted with caution given the large standard deviation seen in that study. On the other hand, Scheuer et al. [27] showed that the binding of near infrared labeled pertuzumab was not affected by the presence of unlabeled trastuzumab. Assuming this data by Marquez et al. were reliable, ^89^Zr-pertuzumab would have limited value in monitoring response to trastuzumab treatment. Here, we showed that anti-EGFR nimotuzumab (domain III) did not alter the in vitro binding (Figure 2) and tumor uptake in vivo (Figure 5 and Appendix A) of anti-EGFR ^89^Zr-8709-scFv-Fc. Similarly, the presence of 8709-scFv-Fc did not alter the binding of ^111^In-nimotuzumab in vitro and in vivo. We also previously showed using near infrared labeled 8709 antibody fragments did not bind cells overexpressing mutant EGFR*vIII* which has all of domain I and most of domain II deleted [17].

Our in vivo studies show a rather low tumor uptake of the ^89^Zr-8709-scFv-Fc imaging agent. The highest tumor uptake was 6.5 %IA/cc by microPET which was almost two-fold less that for ^89^Zr-nimotuzumab seen in this model in a previous study [19]. This rather low tumor uptake may be attributed to the high K_D_ of 8709-scFv-Fc fragment (72 nM) compared with nimotuzumab (14 nM). The pharmacokinetics of ^89^Zr-8709-scFv-Fc was similar to ^89^Zr-nimotuzumab, indicating that it is not the reason for the observed differences in tumor uptake of ^89^Zr-8709-scFv-Fc versus ^89^Zr-nimotuzumab [19]. In addition, we observed a rather higher than normal kidney uptake for ^89^Zr-8709-scFv-Fc. Since the size of scFv-Fc is greater than the kidney cutoff this high uptake maybe suggestive of in vivo proteolysis. Additional studies are needed to investigate the stability of this fragment. Strategies such as affinity maturation could be employed in the future to improve the binding and therefore tumor uptake of ^89^Zr-8709-scFv-Fc.

## 5. Conclusions

Necitumumab is the first anti-EGFR therapeutic antibody that is indicated only in squamous NSCLC patients whose cancer is positive for EGFR by immunohistochemistry [28]. Given the inherent issues associated with EGFR determination by immunohistochemistry, better strategies to diagnose, select patients for anti-EGFR treatments and monitor response are needed. Agents that bind to the same epitope as the therapeutic agent would have limited value, hence the need for those that bind to different epitopes. Here, we presented an anti-EGFR domain II PET imaging agent ^89^Zr-8709-scFv-Fc. The domain specificity of ^89^Zr-8709-scFv-Fc indicates that this immunoPET agent may find applications for diagnosis, patient selection and monitoring of response to anti-EGFR treatments. In its current form, translation of ^89^Zr-8709-scFv-Fc will be limited by the rather low tumor uptake shown in these studies, which is due in part by the high K_D_ (72 nM). Most therapeutic antibodies in this class are pico–low nanomolar affinities. Ongoing engineering studies on affinity maturation of 8709-scFv-Fc/8709-IgG would improve tumor uptake and hence imaging characteristics of the imaging agent(s).

## Figures and Tables

**Figure 1 cancers-13-00560-f001:**
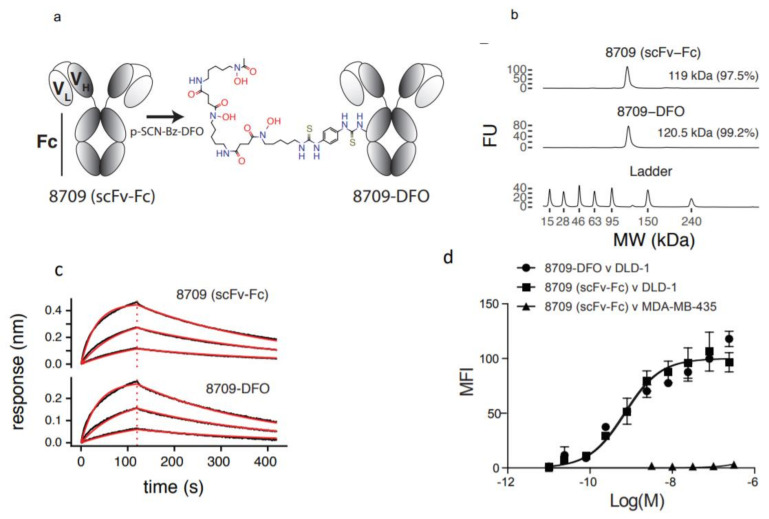
Characterization of DFO-8709-scFv-Fc. (**a**) Conjugation of 8709-scFv-Fc to DFO, (**b**) Bioanalyzer analysis for 8709-scFv-Fc and 8709-scFv-Fc-DFO, Bioanalyzer shows ladder (L), 8709-scFv-Fc, DFO-8709-scFv-Fc (**c**) biolayer interferometry (BLI) (**d**) Flow cytometry analysis of 8709-scFv-Fc and DFO-8709-scFv-Fc in EGFR positive DLD-1 and EGFR negative MDA-MB-435 cell lines.

**Figure 2 cancers-13-00560-f002:**
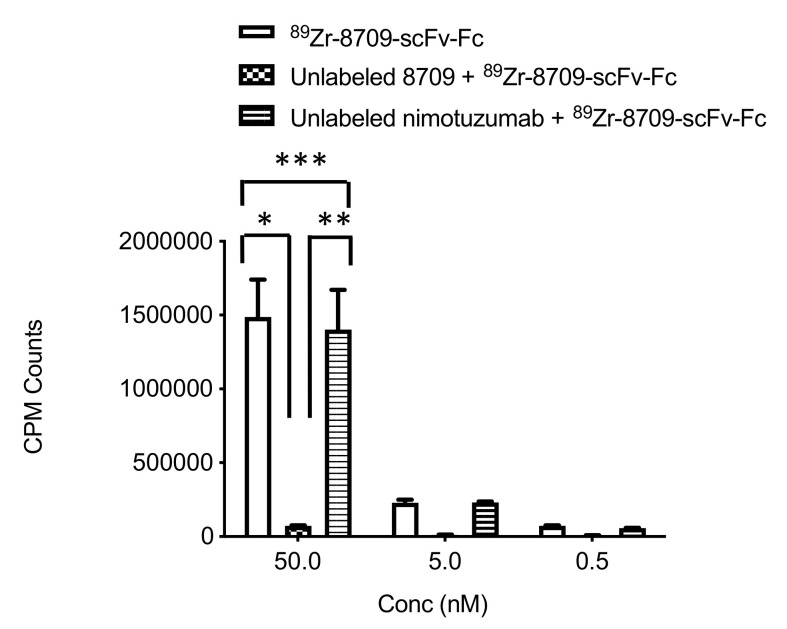
Competitive binding assay of ^89^Zr-8709-scFv-Fc in EGFR positive DLD-1 cells at different concentration 50 nM, 5 nM and 0.5 nM in the presence of unlabeled 8709-scFv-Fc or unlabeled nimotuzumab (domain III binder). *p* < 0.05 for * and ** indicating statistical significance, while *p* > 0.05 for *** at 50 nM concentration, and as well at 5 and 0.5 nM.

**Figure 3 cancers-13-00560-f003:**
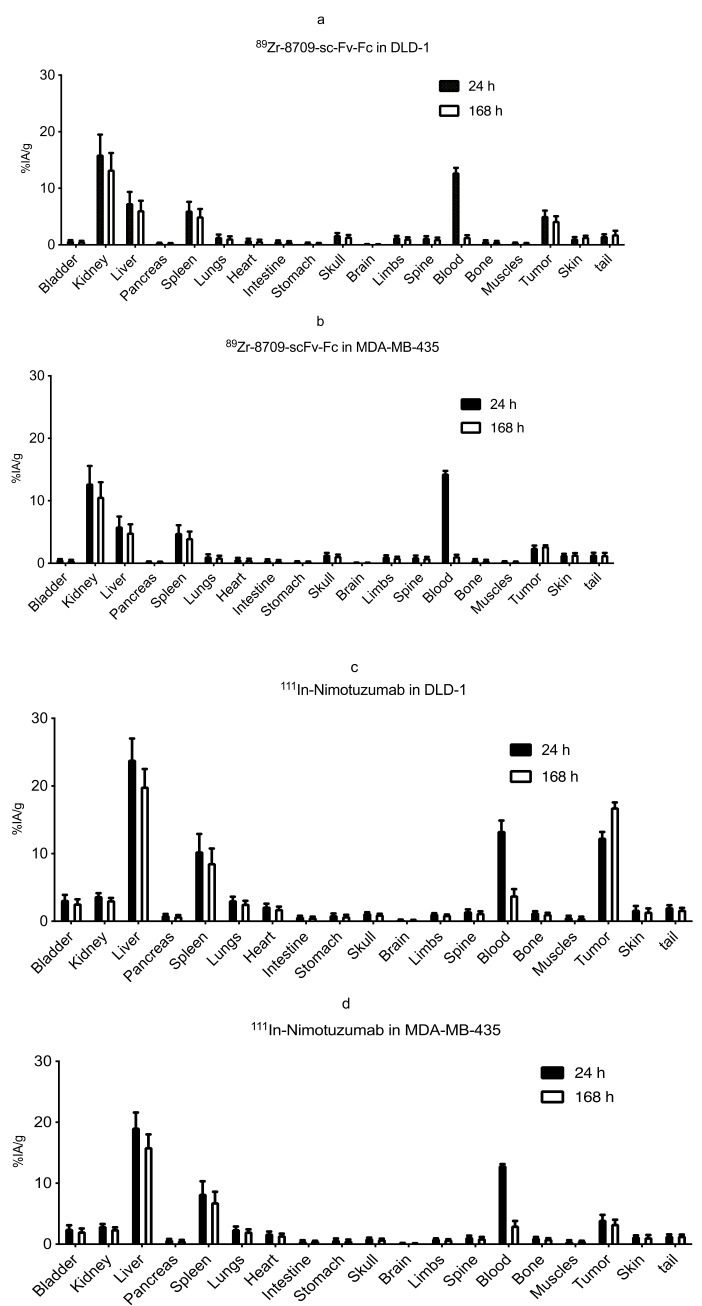
(**a**–**d**). Biodistribution of ^89^Zr-8709-scFv-Fc in athymic CD-1 nude mice bearing EGFR positive xenograft DLD-1 (**a**) and EGFR negative MDA-MB-435 model (**b**) at 24 h and 120 h post injection (p.i.). Biodistribution of ^111^In-nimotuzumab in athymic CD-1 nude mice bearing EGFR positive xenograft DLD-1 (**c**) and EGFR negative MDA-MB-435 model (**d**) at 24 and 120 h p.i. Tissue uptake was expressed as % injected activity per gram (% IA/g) +/− SEM.

**Figure 4 cancers-13-00560-f004:**
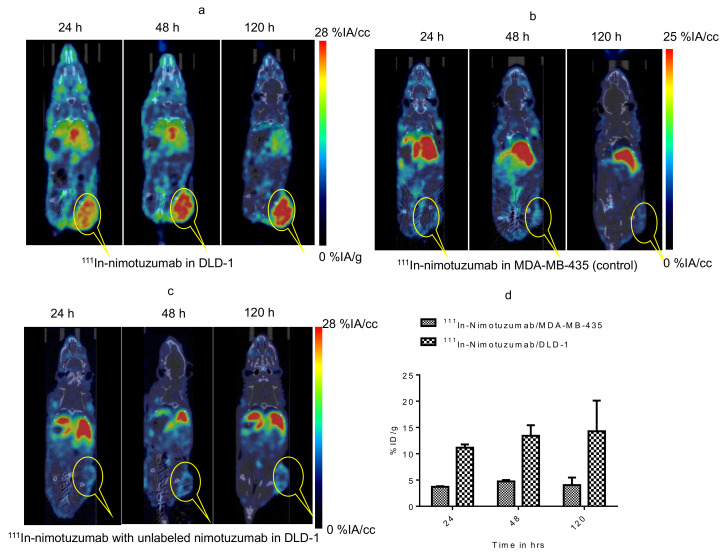
Micro SPCET/CT imaging of Domain III binding of ^111^In-nimotuzumab in CD-1 nude mice bearing (**a**) EGFR positive DLD-1, (**b**) control MDA-MB-435 and (**c**) Pre-blocking with cold nimotuzumab followed by administration of ^111^In-nimotuzumab and at 24, 48, and 120 h p.i. A group of mice was initially pre-injected with 250 µg of unlabeled nimotuzumab to block domain III, and was followed by a tail vein injection of ^111^In-nimotuzumab c (domain III specific). (**d**) Quantification of tumor uptake presented as %IA/g +/− SEM.

**Figure 5 cancers-13-00560-f005:**
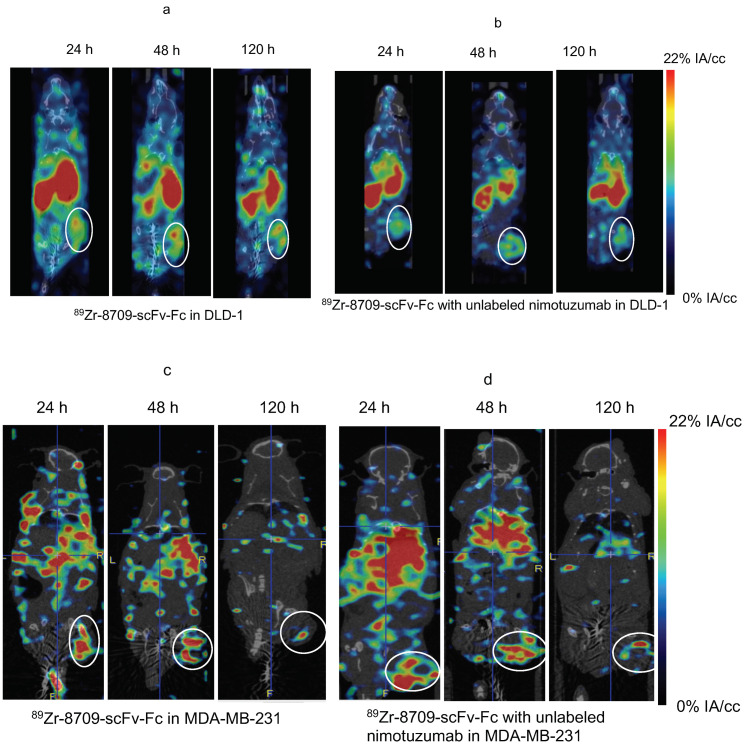
MicroPET imaging of ^89^Zr-8709-scFv-Fc (domain II binder) in CD-1 nude mice bearing (**a**) EGFR positive DLD-1, (**b**) Pre-blocking DLD-1 tumor bearing mice with cold nimotuzumab followed by administration of ^89^Zr-8709-scFv-Fc, (**c**) EGFR positive MDA-MB-231, (**d**) Pre-blocking MDA-MB-231 tumor bearing mice with cold nimotuzumab followed by administration of ^89^Zr-8709-scFv-Fc, (**e**) control MDA-MB-435 xenografts at 24, 48, and 120 h p.i., and (**f**) Quantification of tumor uptake presented as %IA/g +/− SEM. Mice was initially pre-injected with 250 µg of unlabeled nimotuzumab to block domain III, and was followed by a tail vein injection of ^89^Zr-8709-scFv-Fc (domain II specific). * indicates no statistical difference between ^89^Zr-8709-scFv-Fc vs unlabeled nimotuzumab + ^89^Zr-8709-scFv-Fc. ** indicates significant difference between EGFR positive DLD-1 and control MDA-MB-453 xenografts.

**Figure 6 cancers-13-00560-f006:**
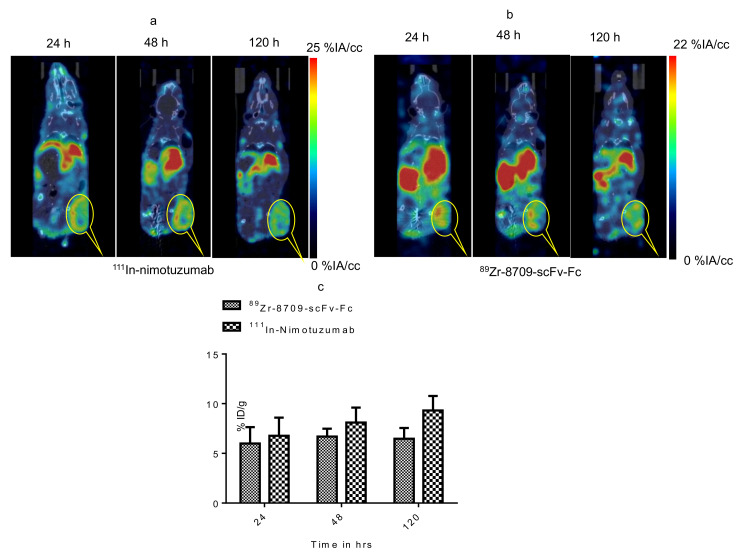
Simultaneous dual tracer (^111^In-nimotuzumab and ^89^Zr-8709-scFv-Fc) imaging of EGFR domain III and domain II in athymic CD-1 nude mice bearing DLD-1 xenograft. (**a**) ^111^In-nimotuzumab window, (**b**) ^89^Zr-8709-scFv-Fc window and (**c**). Bar graph representing the tumor uptake for ^89^Zr-8709-scFv-Fc window and ^111^In-nimotuzumab window.

**Table 1 cancers-13-00560-t001:** Pharmacokinetic parameters of ^89^Zr-8709-scFv-Fc ± SD.

Compound	t_1/2α_ (h)	t_1/2β_ (h)	AUC (% IA.h/mL)	V_1_ (mL)	CL (mL/h)
^89^Zr-8709-scFv-Fc	1.5 ± 1.2	119.2 ± 1.3	846.5 ± 12.9	13.6 ± 0.5	5.6 ± 0.3

## Data Availability

The data presented in this study are available on request from the corresponding author.

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
