# Peer review of "89Zr-Labeled Domain II-Specific scFv-Fc ImmunoPET Probe for Imaging Epidermal Growth Factor Receptor In Vivo"

_cancers, 2021, doi:10.3390/cancers13030560_

Round 1

Reviewer 1 Report

The submission includes the evaluation of the property of 89Zr-labeled 8709 scFv-Fc targeting EGFR domain II as a PET imaging probe for patient selection and monitoring of response to treatment with anti-EGFR antibodies recognizing the domain III such as nimotuzumab. 89Zr-labeled 8709 showed higher uptake in an EGFR-positive tumor than a negative one. Tumor uptake of 89Zr-labeled 8709 did not decrease by adding nimotuzumab in vitro and in vivo. This property is important to monitor response to anti-EGFR therapy. 89Zr-labeled 8709 seems to a good imaging probe; unfortunately, its tumor uptake was quite low, and the probe would not be applicable to clinical use. As mentioned by the authors in the Discussion section, a higher affinity antibody is necessary to achieve the purpose. Therefore, I don't think the submission deserves publication in Cancers because readers in Cancers has no interest in the results. However, the research concept is great and some researchers in nuclear medicine have an interest, so I recommend the authors submit their manuscript to a journal in nuclear medicine. I hope the paper is accepted in such a journal and the authors develop an ideal imaging probe. I provided some specific comments below.

Major points

L311. The tumor uptake of 111In-nimotuzumab is 12.4%IA/g in the main text, whereas, in Figure 2C, the uptake seems approximately 20%IA/g. The authors need to clarify it.

Figure 6C. The tumor uptake of 111In-nimotuzumab is much lower compared with the data in Figure 2C; they are approximately a half. The authors need to clarify that in the main text. In addition, they should add a statistical analysis.

Discussion. The first and second paragraphs need to be omitted or moved to the Introduction section. The description of HER2 imaging in L407-422 needs to be omitted.

Minor points

L72-73. I agree that 89Zr-cetuximab can be used for diagnosis, but not for patient selection. I'm not sure the EGFR expression is related to the efficacy of anti-EGFR therapy. If the authors know that, they should include an appropriate reference.

L282. The SEC-HPLC data should be provided.

L3016-313. Tumor-to-blood ratios need to be included. The ratios of 89Zr-8709 may be higher than those of 111In-nimotuzumab.

Figure 2 needs to include statistics.

Figure 3 is too small.

Figures 4, 5, and 6 need to include color bars of the image window.

Author Response

File attached.

Reviewer 2 Report

Alizadeh and colleagues have provided important data regarding monitoring response to anti-EGFR antibodies. Assessing of EGFR overexpression in quantificational mode by domain II-specific anti EGFR antibody 89Zr-8709-scFv-Fc allows better patient selection and monitoring treatment response. Furthermore, it did not influence binding of Nimotuzumab, both in vitro and in vivo, proved by simultaneously imaging of domain III and domain II via injection of 89Zr-8709-scFv-Fc and 111In-nimotuzumab.
The advantage of the work is developing a domain II-specific antibody fragment 8709, as well as thoroughly carried out and carefully described methods.

I have just a few questions:

1.
Figure 5 shows pre-blocking MDA-MB-358 231 tumor bearing mice with cold nimotuzumab followed by administration of 89Zr-8709-scFv-Fc. How do you interpret decreasing of uptake at 120 h p.i.? Does it reflect the activity of Nimotuzumab 5 days after the treatment exposure?

2.
Line 440 – consider rephasing the sentence.
Necitumumab is not indicated to all cancer’s diagnosis, but for NSCLC with squamous phenotype (NCT01624467 study included also solid tumours). Moreover, the latest meta-analyses showed longer OS in patients with high expression of the EGFR receptor. Expression is usually tested by IHC and Necitumumab binds as other EGFR antibodies to epitope III.
References:
U.S. Food and Drug Administration. FDA approves Portrazza to treat advanced squamous non-small-cell lung cancer. Available online: https://www.fda.gov/ drugs/new-drugs-fda-cders-new-molecular-entitiesand-new-therapeutic-biological-products/novel-drugapprovals-2015.

Wang L, Liao C, Li M, Zhang S, Yi F, Wei Y, Yu J, Zhang W. Necitumumab plus platinum-based chemotherapy versus chemotherapy alone as first-line treatment for stage IV non-small cell lung cancer: a meta-analysis based on randomized controlled trials. Ann Palliat Med. 2020 Sep 15:apm-19-365. doi: 10.21037/apm-19-365. Epub ahead of print. PMID: 32954751.)

3.
In vivo studies show a rather low tumour uptake of the 89Zr-8709-scFv-Fc imaging agent. Do you have any suggestions how the affinity of the antibody can be improved? Moreover, what constraints/limitations do you expect in evaluating this imaging agent in humans?
It would be important for the reader to emphasize the practical significance of the research a little more and to develop the issue of the practical application of the research results.

Author Response

File Attached

Round 2

Reviewer 1 Report

The tumor uptake of 89Zr-labeled 8709 was quite low, and the probe would not apply to clinical use. Unfortunately, I don't think the submission deserves publication in Cancers. It is highly recommended that the authors submit the manuscript to a journal of nuclear medicine. I hope the treatise will be accepted by such a journal.